

# Dissipation rate of turbulent kinetic energy in stably stratified sheared flows

Sergej Zilitinkevich[1,2,3,4], Oleg Druzhinin[5], Andrey Glazunov[6], Evgeny Kadantsev[2], Evgeny Mortikov[4,6], Iryna Repina[4,7], Yulia Troitskaya[5]

[1]Finnish Meteorological Institute, Helsinki, 00101, Finland
[2]Institute for Atmospheric and Earth System Research / Physics, Faculty of Science, University of Helsinki, 00014, Finland
[3]Lobachevsky State University of Nizhni Novgorod, 603950, Russia
[4]Lomonosov Moscow State University, 117192, Russia
[5]Institute of Applied Physics, Russian Academy of Sciences, Nizhny Novgorod, 603950, Russia
[6]Institute of Numerical Mathematics, Russian Academy of Sciences, Moscow, 119991, Russia
[7]Obukhov Institute of Atmospheric Physics, Russian Academy of Sciences, Moscow, 119017, Russia

*Correspondence to*: Sergej Zilitinkevich (sergej.zilitinkevich@fmi.fi)

**Abstract.** Over the years, the problem of dissipation rate of turbulent kinetic energy (TKE) in stable stratification remained unclear because of the practical impossibility to directly measure the process of dissipation that takes place at the smallest scales of turbulent motions. Poor representation of dissipation causes intolerable uncertainties in turbulence-closure theory and, thus, in modelling stably stratified turbulent flows. We obtain theoretical solution to this problem for the whole range of stratifications from neutral to limiting stable; and validate it via (i) direct numerical simulation (DNS) immediately detecting the dissipation rate and (ii) indirect estimates of dissipation rate retrieved via the TKE-budget equation from atmospheric measurements of other components of the TKE-budget. The proposed formulation of dissipation rate will be of use in any turbulence-closure models employing the TKE budget equation and in problems requiring precise knowledge of the high-frequency part of turbulence spectra in atmospheric chemistry, aerosol science and microphysics of clouds.

## 1 Introduction

Until present, dependence of dissipation rate, $\varepsilon_K$, of turbulent kinetic energy (TKE), $E_K$, on static stability remained insufficiently understood that caused principal difficulties in turbulence-energetics/closure theory and intolerable uncertainties in comprehending and modelling stably-stratified turbulent flows. Traditionally, the dissipation rate is parameterised in terms of a turbulent length-scale, $l_T$, as $\varepsilon_K \sim E_K^{3/2}/l_T$. This solves the problem in neutrally stratified boundary-layer flow, when the only length-scale is distance over the surface, $z$, so that $l_T \sim z$. However, in stratified flows, one more length-scale appears, namely the Obukhov length scale, $L$, so that the ratio $l_T/z$ becomes an unknown function of $z/L$. To define this function we combine observational evidence with theoretical analyses. We employ the steady-state TKE budget equation to retrieve data on dissipation versus stability from uncountable data on wind profiles in moderately stable





stratified atmospheric surface layers; supplement this information with our own direct numerical simulation of turbulence in stably stratified Couette flow; and combine the collected empirical knowledge with asymptotic analysis of the TKE equation. The analyses reveals perfect equivalence of our *asymptotic formulation of the velocity profile in extremely stable stratification and well-known log-linear velocity profile in moderately stable stratifications* typical of the atmospheric

surface layer – up to the coincidence of empirical dimensionless constants. This very lucky empirical finding yields universal formulation of dissipation rate versus static stability, valid over the whole range of stratifications from neutral to extremely stable. The formulation is applicable to any stationary and horizontally homogeneous stably stratified sheared flows and can be used within any turbulence-closure model equipped with TKE budget equation.

For certainty, we consider the dissipation rate of TKE in terms of dry atmosphere, where the fluctuation of buoyancy,

$b = \beta\theta$, is proportional to fluctuation of potential temperature, $\theta$; $\beta = g/T_0$ is the buoyancy parameter; $g$ is the gravitational acceleration; and $T_0$ is reference value of absolute temperature. Since Kolmogorov (1942), $\varepsilon_K$ is expressed through the dissipation length-scale $l_T$ or time-scale, $t_T$:

$$\varepsilon_K = \frac{E_K}{t_T} = \frac{E_K^{3/2}}{l_T} \;. \tag{1}$$

This formulation is not hypothetical but just defines the scales $t_T$ and $l_T$, so that Eq. (1) merely expresses one unknown, $\varepsilon_K$,

through another, $t_T$ or $l_T$. In neutrally stratified boundary-layer flows, the only principal length-scale is the height over the surface, $z$; so that $l_T$ is proportional to $z$, which yields

$$l_T = C_l z, \;\; \varepsilon_K = \frac{E_K^{3/2}}{C_l z} \,, \tag{2}$$

where $C_l$ is dimensionless constant to be determined empirically.

Stratification involves the Obukhov length-scale:

$$L = \frac{\tau^{3/2}}{-\beta F_z} \,, \tag{3}$$

where $\tau$ is absolute value of vertical turbulent flux of momentum $\boldsymbol{\tau} = (\tau, 0)$, and $F_z$ is vertical turbulent flux of potential temperature (Obukhov, 1946). The restraining effect of stable stratification on turbulence is characterised by the dimensionless height, $z/L$; gradient Richardson number:

$$\mathrm{Ri} = \frac{\beta \partial\Theta/\partial z}{(\partial\mathbf{U}/\partial z)^2} \;; \tag{4}$$

or flux Richardson number:

$$\mathrm{Ri}_f = \frac{\beta F_z}{\boldsymbol{\tau}\cdot\partial\mathbf{U}/\partial z} \,, \tag{5}$$



where $\partial \mathbf{U}/\partial z$ and $\partial \Theta/\partial z$ are vertical gradients of mean wind velocity, $\mathbf{U} = (U, V)$, and mean potential temperature, $\Theta$. Then the dimensionless dissipation rate, $\varepsilon_K z/E_K^{3/2}$, is no longer a constant but depends on stratification $(z/L, \text{ Ri or Ri}_f)$. Until recently, practically nothing was known about this dependence beyond the interval of stratifications covered by observations in atmospheric surface layer: $0 < \text{Ri} < 0.2$, which corresponds to $0 < z/L < 10$.

## 2 Dissipation rate in steady-state, stably stratified sheared flows

We consider horizontally homogeneous stationary boundary-layer flow in semi-space $z > 0$ as idealised model of atmospheric surface layer. Here, the familiar TKE budget equation expresses the dissipation rate, $\varepsilon_K$, through $\tau$, $F_z$ and $\partial U/\partial z$:

$$\varepsilon_K = \tau \frac{\partial U}{\partial z} + \beta F_z = \tau \frac{\partial U}{\partial z}\left(1 - \text{Ri}_f\right). \tag{6}$$

With increasing static stability, $\text{Ri}_f$ obviously increases but (because $\varepsilon_T > 0$) remains limited, which is why it must tend to a finite limit: $\text{Ri}_f \to R_\infty < 1$. Atmospheric data and results from direct numerical simulation (DNS) demonstrated below in Figures 2 and 3 confirm such behaviour and yield quite certain estimate of $R_\infty = 0.2$.

Then, substituting $R_\infty$ for $\text{Ri}_f$ in Eq. (5) yields asymptotic expression of the velocity gradient in extremely stable stratification:

$$\frac{\partial U}{\partial z} \to \frac{1}{R_\infty} \frac{\tau^{1/2}}{L} \equiv \frac{\tau^{1/2}}{L}\left(\frac{k}{R_\infty} \frac{z}{L}\right) \text{ at } \frac{z}{L} \to \infty . \tag{7}$$

Here, von Karman constant, $k$, is inserted in numerator and denominator just to highlight consistency of Eq. (7) with well-known Monin-Obukhov Similarity Theory (MOST) by Monin and Obukhov (1954) formulation of the velocity-gradient at weakly and moderately stable stratifications typical of atmospheric surface layers:

$$\frac{\partial U}{\partial z} = \frac{\tau^{1/2}}{kz}\left(1 + C_u \frac{z}{L}\right), \tag{8}$$

where $C_u = 2$ is well-established dimensionless empirical constant (e.g., Monin and Yaglom, 1971, Garratt, 1992, Stull, 1997). Originally, Eq. (8) was derived as the first term in the Taylor expansion of dimensionless velocity gradient, $\Phi_M = \left(kz/\tau^{1/2}\right)\partial U/\partial z$, considered in MOST as universal function of $z/L$. Subsequently, it was revealed that Eq. (8) with $C_u = 2$ is valid over the whole range of $z/L$ observed in atmospheric surface layers, $0 < z/L < 10$, which corresponds to quite small gradient Richardson numbers: $0 < \text{Ri} < 0.2$ (Monin and Yaglom, 1971). By this means, Eq. (8) with $C_u = 2$ based on massive atmospheric data for moderately stable stratifications yields at $z/L \to \infty$ precisely the same limit as Eq. (7) with $k/R_\infty = 2$ obtained from conventional value of $k = 0.4$ and new estimate of critical flux Richardson number $R_\infty = 0.2$ obtained from topical DNS and selected atmospheric data for unusually stable stratifications (Figure 2). This lucky coincidence just means that Eq. (8) with $C_u = k/R_\infty$ hold true in any stable stratification:



$$\frac{\partial U}{\partial z} = \frac{\tau^{1/2}}{kz}\left(1 + \frac{k}{R_\infty}\frac{z}{L}\right) \text{ at } 0 < z/L < \infty \,. \tag{9}$$

Then, substituting Eq. (9) for $\partial U/\partial z$ into Eqs. (5) and (6), yields simple relations linking $\mathrm{Ri}_f$ with $z/L$:

$$\mathrm{Ri}_f = \frac{kz/L}{1 + kR_\infty^{-1}z/L}, \quad \frac{z}{L} = \frac{R_\infty}{k}\frac{\mathrm{Ri}_f}{R_\infty - \mathrm{Ri}_f}, \tag{10}$$

and exact formulation of the TKE dissipation rate as dependent on static stability:

$$5 \quad \varepsilon_K = \frac{\tau^{3/2}}{kz}\left[1 + k(R_\infty^{-1} - 1)\frac{z}{L}\right] = \frac{\tau^{3/2}}{kz}\frac{1 - \mathrm{Ri}_f}{1 - \mathrm{Ri}_f/R_\infty}\,. \tag{11}$$

It is worth noting that $R_\infty$ can be derived from well-established phenomenological constants of turbulence in the inertial subrange (Katul et al., 2014). The actual value in this case is slightly higher ($R_\infty = 0.25$) but still within reasonable range.

To comprehensively validate the above analyses, we performed direct numerical simulation (DNS) of stably stratified *Couette flow*, namely, the plain-parallel flow between two horizontal plates separated in the vertical by distance $d$, and

moving with constant velocity in opposite directions. To assure accuracy of numerical simulations, we employed two different codes: INM-RAS (Mortikov, 2016), and IAP-RAS (Druzhinin et al., 2016) that have shown quite consistent results. In our DNS total (turbulent + molecular) fluxes of momentum, $\tau$, and potential temperature, $F_z$, are practically equal to the turbulent fluxes elsewhere beyond narrow near-wall sublayers where molecular transports dominate. In Couette flow these fluxes are constant with height, as in surface-layer flows. Similarly, flux-profile relations linking $\tau$ and $F_z$ with vertical

gradients of mean velocity, $\partial U/\partial z$, and potential temperature, $\partial\Theta/\partial z$, as well as the budget equations for turbulent energies, in particular Eq. (6), are the same as in surface layer flows. The only difference is in geometry of domains illustrated in Figure 1.

Following Obukhov (1942), we distinguish between "absolute geometry" characterised by usual height over the surface, $z$, and "internal geometry" characterised in Couette flow by specific vertical coordinate, $\tilde{z}$, dictated by conformal mapping of

the Couette-flow domain ($0 < z < d$) into semi-space:

$$\tilde{z} = \frac{d}{\pi}\sin\frac{\pi z}{d} \text{ in Couette flow } (0 < z < d) \,. \tag{12}$$

This coordinate reflects equal influences of lower and upper walls on a fluid flow.

In semi-space, the "internal geometry" coincides with "absolute geometry": $\tilde{z} = z$. Thus, vertical structure of Couette flow in terms of $\tilde{z}$ coincides with vertical structure of the surface-layer flow in terms of $z$, which allows showing in the same

framework the genuine dissipation rate: $\varepsilon_K = \nu\langle(\partial u_i/\partial x_k)(\partial u_i/\partial x_k)\rangle$, where $\nu$ is kinematic viscosity, calculated from DNS together with $\varepsilon_K = \tau\,\partial U/\partial z + \beta F_z$ retrieved from atmospheric observations assuming the steady-state TKE budget.

In Figures 2-4 we show our DNS data together with data from observations in atmospheric surface layer provided the estimates of dissipation rate (Figure 3):

•   via the Kolmogorov −5/3 power law from measured spectra of TKE in the inertial subrange (Pearson et al., 2002), and



- via the steady-state TKE budget Eq. (6) from measured turbulent fluxes of momentum, $\tau$, and potential temperature, $F_z$, together with vertical gradient of wind velocity, $\partial U/\partial z$.

In these figures DNS data are shown by heavy coloured dots; and atmospheric data, by light grey symbols.

Figure 2 shows flux Richardson number, $\mathrm{Ri}_f = \beta F_z (\boldsymbol{\tau} \cdot \partial \mathbf{U}/\partial z)^{-1}$, versus dimensionless height, $\tilde{z}/L$ in Couette flow or $z/L$ in atmospheric surface layer. Black curve is plotted after Eq. (10) taking conventional value of von Karman constant: $k = 0.4$ and our estimate of the maximal flux Richardson number: $R_\infty = 0.2$ resulted from best fit of Eq. (10) to DNS data. Notably, total (turbulent + molecular) fluxes of momentum, $\tau$, and potential temperature, $F_z$, in Couette flow are constant across the flow which assures very certain specification of $\mathrm{Ri}_f$ and $L$, and makes our DNS most suitable for calibrating the theory. We recall that Eqs. (10) and (11) are relevant to the well-developed turbulence regime where molecular transports are negligible, so that turbulent fluxes practically coincide with total fluxes. In our DNS, it is so except for narrow transition layers dominated by molecular transport near the lower and upper walls: $0 < \tilde{z} < 50\nu/\tau^{1/2}$. Data from these layers are indicated by dark grey points. Light grey symbols show atmospheric data from the following sources: research observatory Tiksi in East Siberia (Grachev et al., 2018) near the Arctic Ocean coast; offshore oceanographic platform in the Black Sea (Repina et al., 2009); and acoustic soundings over arid-steppe in Republic of Kalmykia in Southern Russia (Vazaeva et al., 2017). In spite of inevitable heterogeneity, non-stationarity and other side effects, atmospheric data correlate quite well with DNS data.

Figure 3 shows dimensionless dissipation rate, $\varepsilon_K z/\tau^{3/2}$, versus $z/L$ after Eq. (11) and atmospheric data, and $\varepsilon_K \tilde{z}/\tau^{3/2}$ versus $\tilde{z}/L$ after DNS in Couette flow. All notations are the same as in Figure 2. The theoretical curve plotted after Eq. (11) with $k = 0.4$ and $R_\infty = 0.2$ is fully consistent with experimental data except for narrow transition layer $0 < \tilde{z} < 50\nu/\tau^{1/2}$, where Eq. (11) is irrelevant. Hence, Figure 3 justifies the stability dependence of dissipation rate, Eq. (11), and provides additional confirmation to empirical estimate of $R_\infty = 0.2$.

In Figure 2 we use as argument $\mathrm{Ri}_f = \frac{\beta F_z}{\boldsymbol{\tau} \cdot \partial \mathbf{U}/\partial z}$, where turbulent fluxes (disregarding molecular contributions in the transition layer) appear in both numerator and denominator. Hence uncertainties in both fluxes are somehow compensated. This is not the case in Figure 3 ($\varepsilon_K z/\tau^{3/2}$ vs. $z/L$): the dissipation rate in the numerator is just total dissipation, whereas the momentum flux in denominator disregards the molecular contribution. This just causes ugly looking (but only natural) dark gray points on the left side of Figure 3.

## 3 Turbulent length-scales and general criterion of stratification

The concept of TKE dissipation rate is directly related to definition of turbulent time-scale, $t_T \equiv E_k/\varepsilon_K$, and length-scale, $l_T \equiv E_K^{1/2} t_T = E_K^{3/2}/\varepsilon_K$. Then Eq. (11) defines $l_T$ as function of $z/L$:

$$l_T \equiv E_K^{1/2} t_T = \frac{E_K^{3/2}}{\varepsilon_K} = kz \left(\frac{E_K}{\tau}\right)^{3/2} \left[1 + k(R_\infty^{-1} - 1)\frac{z}{L}\right]^{-1}. \tag{13}$$



It has asymptotic limits:

$$l_T \to k(E_K/\tau)_0^{3/2} z \sim z \ \text{ at } \ z/L \to 0 \ \text{ and } \ l_T \to \frac{R_\infty}{1-R_\infty}\left(\frac{E_K}{\tau}\right)_\infty^{3/2} L \sim L \ \text{ at } \ z/L \to \infty \,, \tag{14}$$

where the limits of $E_K/\tau$ in neutral stratification and extremely stable stratification are dimensionless constants [current estimates based on our DNS: $(E_K/\tau)_0 \approx 4$ and $(E_K/\tau)_\infty \approx 11$]. The length-scale similar to Eq. (13) was already revealed as inherent to spectra of turbulence in unstably stratified boundary-layer flows (Glazunov, 2014).

We emphasise that $l_T$ is the scalar characterising turbulence as a whole. Contrastingly, turbulent mixing in different directions is characterised by the mixing-lengths vector $l_{Ti} \equiv E_{Ki}^{1/2} t_T$ $(i = 1, 2, 3)$ with generally different stream-wise $(i = 1)$, transverse $(i = 2)$ and vertical $(i = 3)$ components. We emphasise principal difference between scalar *length-scale* and vector *mixing-length*. In literature, the words "turbulent length-scale" and "turbulent mixing-length" are often used as interchangeable. This cause intolerable confusion as different components of the mixing length differently depend on static stability (Zilitinkevich et al., 2013).

The above analyses are done for the simplest surface-layer (or Couette) flow, where dimensionless height $z/L$ (or $\tilde{z}/L$) plays the role of criterion quantifying the effect of stratification on turbulence. Luckily, our major result [Eqs. (11) and (13)] is extended to a wide range of stratified turbulent flows. We recall that stratified turbulence is characterised, besides $E_K$, by turbulent potential energy (TPE), $E_P = \frac{1}{2}\beta\langle\theta^2\rangle/\partial\Theta/\partial z$; and quantify the effect of stratification on turbulence by the "energy Richardson number" defined as

$$\mathrm{Ri}_E = \frac{E_P}{E_K} \,. \tag{15}$$

In contrast to traditional criteria, such as Ri (Eq. 4), $\mathrm{Ri}_f$ (Eq. 5), or $z/L$, the energy Richardson number criterion is valid in heterogeneous and non-stationary flows, for any mechanisms of generation of turbulence (including breaking waves, oscillating grid, etc.) and in flows with complex geometry.

Expressing the dissipation rates of TKE and TPE in the steady state through the dissipation time scale, $l_T \equiv E_K^{3/2}/\varepsilon_K$, the budget equations for TKE and TPE become

$$E_K = t_T(-\boldsymbol{\tau}\cdot\partial\mathbf{U}/\partial z + \beta F_z) \,, \tag{16}$$

$$E_P = \frac{1}{2}\frac{\beta\langle\theta^2\rangle}{\partial\Theta/\partial z} = -C_P t_T \beta F_z \,, \tag{17}$$

where $C_P$ is dimensionless universal constant quantifying the difference between dissipation rates of TKE and TPE (Zilitinkevich et al., 2013). Equations (16) and (17) in combination with Eq. (10) yield the following relations linking $\mathrm{Ri}_E$ with $\mathrm{Ri}_f$ or $z/L$:

$$\mathrm{Ri}_E = \frac{C_P}{\mathrm{Ri}_f^{-1}-1} = \frac{C_P kz/L}{1+(R_\infty^{-1}-1)kz/L} \,. \tag{18}$$





Figure 4 shows $Ri_E$ versus $z/L$ or $\tilde{z}/L$ (like in previous figures) after our DNS and atmospheric observations. Theoretical curve is plotted after Eq. (18) taking $k = 0.4$, $R_\infty = 0.2$ and empirical estimate of dimensionless constant $C_P = 0.62$ just obtained from the best fit of Eq. (18) to DNS data. Experimental data clearly demonstrate asymptotic limit:

$$Ri_E \to R_{E\infty} = \frac{C_p}{R_\infty^{-1} - 1} = 0.155 \ \text{ at } \ z/L \to \infty \,, \tag{19}$$

Then, using Eq. (18) to express $z/L$ through $Ri_E$, Eq. (11) in terms of $Ri_E$ becomes:

$$\varepsilon_K = \varepsilon_{K(neutral)} \left(1 - \frac{Ri_E}{R_{E\infty}}\right)^{-1} , \tag{20}$$

where $\varepsilon_{K(neutral)}$ is dissipation rate in neutral stratification. In the surface layer $\varepsilon_{K(neutral)} = \tau^{3/2}/kz$; but generally $\varepsilon_{K(neutral)}$ depends on concrete energy-generation mechanisms and geometry of flow.

There is essential advantage of $Ri_E$ as criterion of stratification in numerical modelling. Turbulent fluxes are usually calculated through the diagnostic down-gradient formulations: $\boldsymbol{\tau} = -K_M \, \partial \mathbf{U}/\partial z$ and $F_z = -K_H \, \partial \Theta/\partial z$, where $K_M$ is eddy viscosity and $K_H$ is eddy conductivity. Then, finite-difference approximation of the gradients causes uncertainties in $\boldsymbol{\tau}$, $F_z$ and, hence, the Obukhov length, $L$ [Eq. (3)], flux Richardson number, $Ri_f$ [Eq. (5)], and gradient Richardson number, $Ri$ [Eq. (4)]. Contrastingly, TKE and TPE are defined from the prognostic budget equations accounting for turbulent diffusion that smooths the energies and assures quite certain calculation of $Ri_E$.

## 4 Concluding remarks

The dissipation rate of TKE, $\varepsilon_K$, as dependent on static stability over years remained uncertain because of impossibility of direct measurement of $\varepsilon_K$. Admittedly, $\varepsilon_K$ can be retrieved via the TKE budget equation from the measured turbulent fluxes, $\tau$ and $F_z$, and mean-velocity gradient, $\partial U/\partial z$, and also via the Kolmogorov $-5/3$ power law from the measured spectra of TKE in the inertial subrange. However, these methods are justified only in stationary and horizontally homogeneous flows and require fully controlled conditions. These necessities, practically unachievable in atmospheric experiments, make estimates of $\varepsilon_K$ from atmospheric observations rather uncertain. Wide spread of atmospheric data is clearly seen in our figures. Moreover, available atmospheric data cover only weakly to moderately stable stratifications typical of the surface layer. To avoid these difficulties, we performed topical DNS of the steady-state, stably stratified turbulent Couette flows up to the strongest attainable stratifications; combined direct data from DNS with data retrieved from atmospheric observations; and employed theoretical analysis to reveal asymptotic behaviour of the mean velocity gradient and dissipation rate in extremely stable stratification: at $z/L \to \infty$, where $L$ is the Obukhov length-scale.

By providential coincidence, the formulations happen to be precisely the same in asymptotic limit $z/L \to \infty$ and in weakly stable stratifications $0 < z/L < 10$ typical of atmospheric surface layer. This yields simple analytical formulations of



dimensionless velocity gradient, $(kz/\tau^{1/2})\,\partial U/\partial z$, and dissipation rate, $(kz/\tau^{3/2})\varepsilon_K$, as universal functions of $z/L$ [Eq. (9) and Eq. (11)] across the whole range of stratifications from neutral to extremely stable.

Universal analytical formulation of $(kz/\tau^{1/2})\,\partial U/\partial z$ versus $z/L$ yields the single-valued relations linking $z/L$ as criterion of stratification in the surface-layer flow or $\tilde{z}/L$ as the same criterion in Couette flow with alternative criterions: flux

Richardson number, $\mathrm{Ri}_f$ [Eq. (5)], and the newly introduced "energy Richardson number", $\mathrm{Ri}_E$ [Eq. (13)], applicable to any turbulent regimes. This opens prospects for extending the obtained dependence of dissipation rate on static stability to any stably stratified turbulent flows.

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

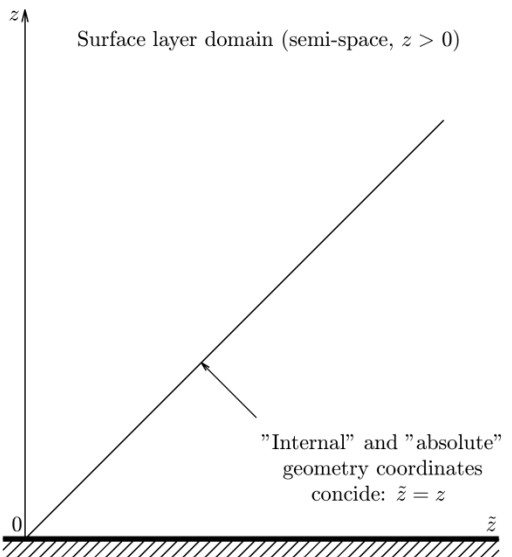
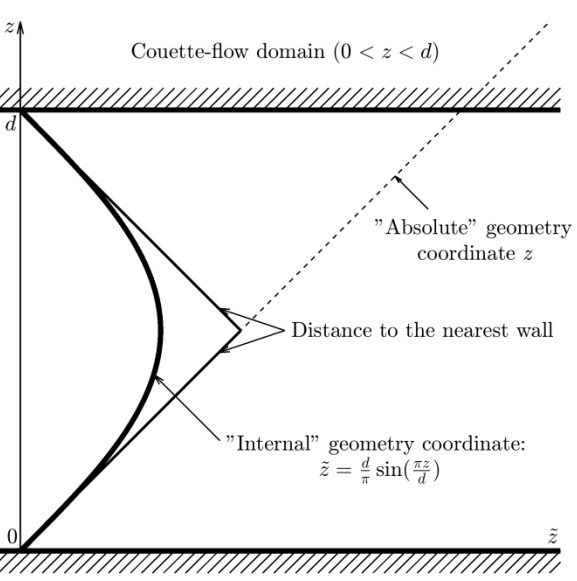

**Figure 1: Usual height, *z*, and vertical coordinate, *z̃*, defined by Eq. (12) characterising "absolute" and "internal" geometry of the domain, respectively. Left panel shows semi-space, *z* > 0, where *z̃* = *z*. Right panel shows the layer between two horizontal walls, 0 < *z* < *d*, where *z̃* coincides with the distance from nearest surface in its close vicinity, but essentially depends on the distances from both surfaces in central part of the domain.**





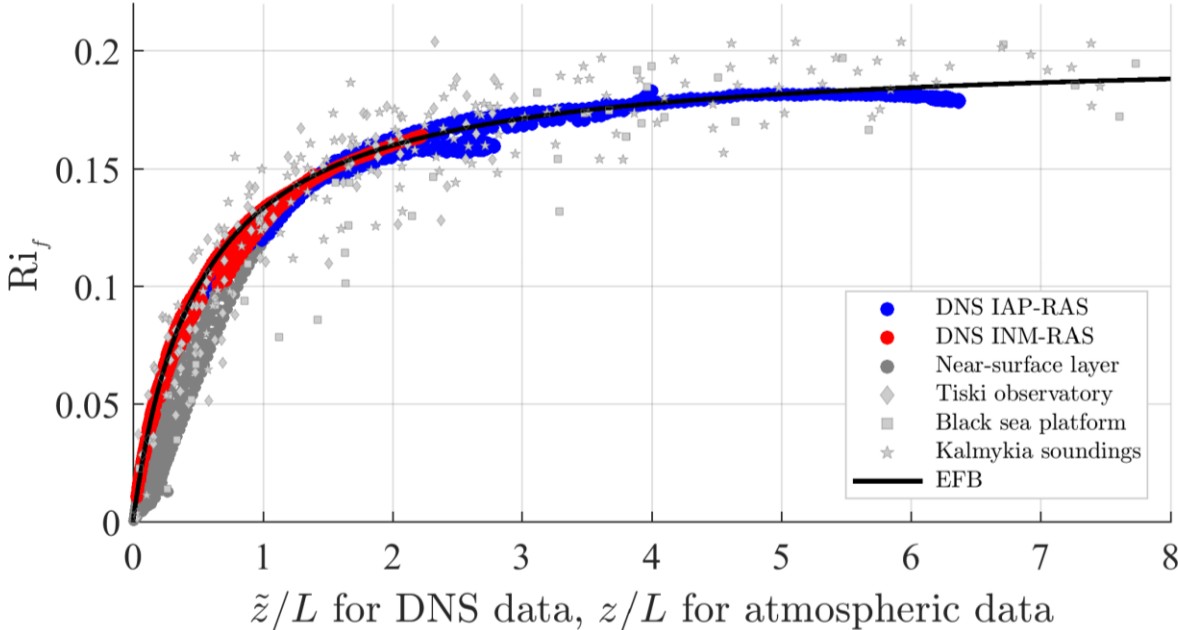

**Figure 2: Flux Richardson number, $Ri_f$, in stable stratification versus $\tilde{z}/L$ in Couette flow or versus $z/L$ in atmospheric surface layer. Empirical data used for the calibration are obtained in two series of DNS runs employing INM-RAS code (red dots) and IAP-RAS code (blue dots). Atmospheric data are taken from Arctic coastal observatory Tiski (light grey diamonds), Black sea offshore platform (light grey squares) acoustic soundings in Kalmykia steppe (light grey stars). Dark grey dots belong to very narrow near-surface layer: $0 < \tilde{z} < 50\nu/\tau^{1/2}$. Black solid line shows Eq. (11) with conventional value of von-Karman constant, $k = 0.4$, and new empirical vale of $R_\infty = 0.2$ just obtained from the best fit of Eq. (10) to DNS data from elsewhere beyond the layer $0 < \tilde{z} < 50\nu/\tau^{1/2}$, where molecular transports are significant and Eq. (10) is not necessarily relevant.**





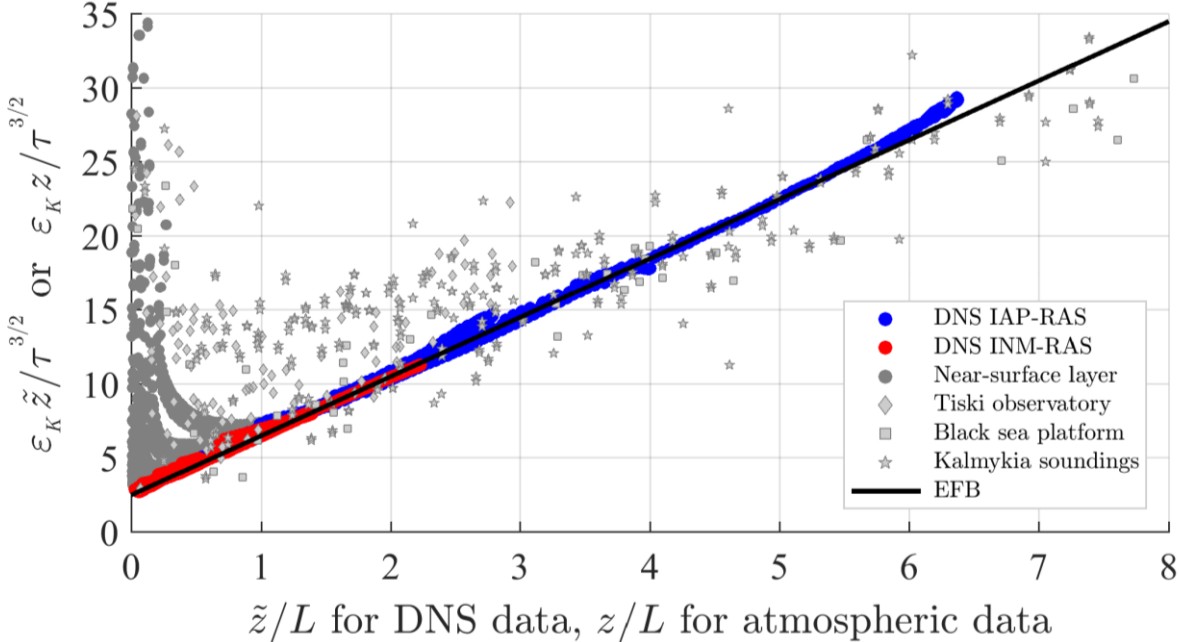

**Figure 3: Dimensionless dissipation rate in stable stratification $\varepsilon_K \tilde{z}/\tau^{3/2}$ versus $\tilde{z}/L$ in Couette flow or $\varepsilon_K z/\tau^{3/2}$ versus $z/L$ in atmospheric surface layer. Empirical data are from the same sources as in Figure 2. Black solid line shows Eq. (11) with $k = 0.4$ and $R_\infty = 0.2$. Dark grey dots belong to very narrow near-surface layer: $0 < \tilde{z} < 50\nu/\tau^{1/2}$, where molecular transports are significant and Eq. (11) is not relevant.**



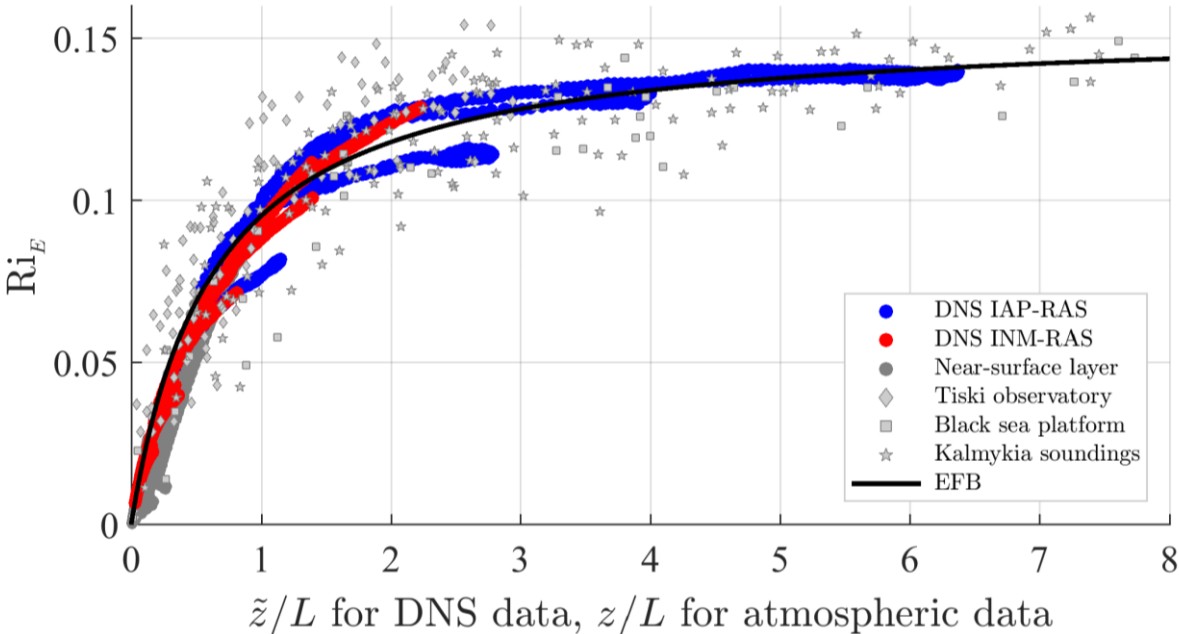

**Figure 4: Energy Richardson number, $Ri_E = E_p/E_K$, versus $\tilde{z}/L$ in Couette flow or versus $z/L$ in atmospheric surface layer. Empirical data are from the same sources as in Figures 2 and 3. Black solid line shows Eq. (16) with $k = 0.4$, $R_\infty = 0.2$, and $C_P = 0.62$.**