# Peer review of "Dissipation rate of turbulent kinetic energy in stably stratified sheared flows"

_Atmospheric Chemistry and Physics, 2018_

## Referee Comment (RC1) · Anonymous Referee #2 · 3 Dec 2018

Review of "*Dissipation rate of turbulent kinetic energy in stably stratified sheared flows*" by Zilitinkevich et al. (MS# ACP-2018-978)

**General Comments**:

The manuscript explores scaling relations for the mean turbulent kinetic energy dissipation rate in a stationary and planar homogeneous stably stratified atmospheric flow. The motivation for the work is that uncertainty in the mean turbulent kinetic energy dissipation rate causes non-trivial uncertainties in closure modeling of stratified atmospheric flow properties, especially relaxation time scales needed in numerous closure schemes and numerical simulations. The analysis is elegant and easy to follow, and the results are insightful. The outcome of the scaling analysis is supported by both field measurements and direct numerical simulations (DNS) of a stratified Couette flow. All in all, the work certainly warrants publication with minor revisions in Atmospheric Chemistry and Physics.

**Minor Comments:**

- p.1, Line 14: "Over years the problem" should be "Over the years, the problem of …"
- p.1, Line 15: " the process of dissipation which takes place" should be "….that takes place"
- p.2, line 1: drop 'topical'
- p.2, line 11 g is the gravitational acceleration, reads better.
- p.2, line 22 – there are font inconsistencies in the stability parameter usage. For example L, the Obukhov length, is capital whereas z/l is used throughout – it should be z/L. Same issue on p.3, line 2.

-The coefficient $C_u$ in equation (8).
According to the Kansas experiment, the stability correction function applied to the mean velocity gradient for **unstable** conditions is $\phi_m = \left(1 - 16\frac{z}{L}\right)^{1/4}$.
If the stability correction function for **stable** conditions is expressed as equation 8, $\phi_m = 1 + C_u\frac{z}{L}$, then continuity of $\phi_m$ is guaranteed as the flow transitions from unstable to stable and vis-a-versa around $\frac{z}{L} = 0$. However, the Kansas experiment did suggest that $\phi_m$ is not only **continuous** but also **smooth** around $\frac{z}{L} = 0$. That is, for small $|\frac{z}{L}|$, the unstable ($\frac{z}{L} < 0$) side leads to $\phi_m = \left(1 - 16\frac{z}{L}\right)^{1/4} \approx 1 - 4\frac{z}{L}$. On the stable side, $1 + C_u\frac{z}{L}$ remains valid for small $|\frac{z}{L}|$. Hence, the Kansas data as well as the continuity condition on $\phi_m$ leads to a $C_u = 4$ not 2. Please comment.

-The value of $R_\infty$: It was shown elsewhere (e.g. Katul et al., 2014) that
$$R_\infty = \frac{1}{1 + \frac{1}{A_\pi}\frac{C_T}{C_o}},$$
where $A_\pi = 1 - 3/5$ is a constant linked to the isotropization of the production term (fast part) correcting the original Rotta model (slow part) and is derived from Rapid Distortion Theory (RDT) in homogeneous turbulence, $C_T = 0.8$ is the Kolmogorov-Obukhov-Corrsin constant associated with the temperature spectrum in the inertial subrange, and $C_o = 0.65$ is the Kolmogorov constant associated

with the vertical velocity energy spectrum within the inertial subrange. Inserting those accepted constants yields $R_\infty = 0.25$, slightly higher than 0.2 (but still within reasonable range). So, the comment here is a suggestion: It is worth noting that $R_\infty$ can be derived from well-established phenomenological constants of turbulence in the inertial subrange.

Page 4, lines 27-28: It is worth showing a 1:1 comparison of the mean turbulent kinetic energy dissipation rate estimates from the spectrum and from the residual of the TKE budget. This additional figure is valuable because it allows an independent 'diagnostic' of how well the assumptions of stationary and planar homogeneous flow in the absence of subsidence and other flux transport terms manifest themselves as errors in the TKE budget assumptions used here.

Page 6, Equation (20) is really the main result as it shows how the turbulent potential energy and the turbulent kinetic energy play a role in shaping the mean turbulent kinetic energy dissipation rate with stability. May be worth expanding this connection in the conclusion.

Figure 3 – worth adding the best-fit line from the Kansas data as well. After all, the TKE budget used here leads to:

$$\varepsilon = \frac{u_*^3}{k_v z}\left[\phi_m\left(\frac{z}{L}\right) - \frac{z}{L}\right] \Rightarrow \frac{\varepsilon\,z}{u_*^3} = \frac{1}{k_v}\left[1 + (C_u + 1)\frac{z}{-L}\right];$$

where $C_u = 4$.

---

## Author Comment (AC1) · 10 Dec 2018

**Response to Anonymous Referee #2**

We are very grateful to reviewer for careful and friendly review of our paper. All the comments are very useful and helped us to improve the manuscript.

The following corrections were made:

*P.1, Line 14: "Over years the problem" should be "Over the years, the problem of …"*

Corrected.

*P.1, Line 15: " the process of dissipation which takes place" should be "….that takes place".*

Corrected.

*P.2, line 1: drop 'topical'.*

Done.

*P.2, line 11 g is the gravitational acceleration, reads better.*

Corrected.

*P.2, line 22 – there are inconsistencies in the definition of the stability parameter. L, the Obukhov length, is capital whereas z/l is used throughout – it should be z/L. Same issue on p.3, line 2.*

Corrected.

*The coefficient Cu in equation (8) …*

The difference is due to definition of the Obukhov length-scale. We define it as $L = \frac{\tau^{3/2}}{-\beta F_z}$, while in papers on the Kansas experiment it is defined including the von Karman constant: $L = \frac{\tau^{3/2}}{-\beta F_z k}$. This just makes approximately two times difference in empirical constants.

*The value of $R_\infty$ …*

P.2, line 6, the following sentences were added:

It is worth noting that $R_\infty$ can be derived from well-established phenomenological constants of turbulence in the inertial subrange (Katul et al., 2014). The actual value in this case is slightly higher ($R_\infty = 0.25$) but still within reasonable range.

*Page 4, lines 27-28: I think it is worth showing a 1:1 comparison of the mean turbulent kinetic energy dissipation rate estimates from the spectrum and from the residual of the TKE budget.*

Such comparison surely deserves consideration. Different types of data used in our paper show very good correspondence between the dissipation rates estimated (i) from spectrum and (ii) as the residual of TKE budget. The difference is within 5% (see the figure below).

[Figure]

In this paper we relay basically on DNS and use atmospheric data only to illustrate principal agreement between surface-layer data and DNS data. We would not like to include additional figures for the following reason. Additional (and not very necessary) figures inevitably diverge the attention of readers from the basic subject.

***Page 6, Equation (20) is really the main result as it shows how the turbulent potential energy and the turbulent kinetic energy play a role in shaping the mean turbulent kinetic energy dissipation rate with stability. May be worth expanding this connection in the conclusion***

Thank you for this valuable remark. We add the following sentences.

1) In the very end of section 3:

There is essential advantage of $\mathrm{Ri}_E$ as criterion of stratification in numerical modelling. Turbulent fluxes are usually calculated through the diagnostic down-gradient formulations: $\boldsymbol{\tau} = -K_M\,\partial\mathbf{U}/\partial z$ and $F_z = -K_H\,\partial\Theta/\partial z$, where $K_M$ is eddy viscosity and $K_H$ is eddy conductivity. Then, finite-difference approximation of the gradients causes uncertainties in $\boldsymbol{\tau}$, $F_z$ and, hence, the Obukhov length, $L$ [Eq. (3)], flux Richardson number, $\mathrm{Ri}_f$ [Eq. (5)], and gradient Richardson number, $\mathrm{Ri}$ [Eq. (4)]. Contrastingly, TKE and TPE are defined from the prognostic budget equations accounting for turbulent diffusion that smooths the energies and assures quite certain calculation of $\mathrm{Ri}_E$.

2) In the very end of concluding remarks:

Universal analytical formulation of $\left(kz/\tau^{1/2}\right)\partial U/\partial z$ versus $z/L$ yields the single-valued relations linking $z/L$ as criterion of stratification in the surface-layer flow or $\tilde{z}/L$ as the same criterion in Couette flow with alternative criterions: flux Richardson number, $\mathrm{Ri}_f$ [Eq. (5)], and the newly introduced "energy Richardson number", $\mathrm{Ri}_E$ [Eq. (13)], applicable to any turbulent regimes. This opens prospects for extending the obtained dependence of dissipation rate on static stability to any stably stratified turbulent flows.

***Figure 3 – worth adding the best-fit line from the Kansas data as well.***

Same reasoning applies here as in comment on comparison of dissipation rate estimated from spectrum and as the residual of TKE budget: in this paper we relay basically on DNS and use atmospheric data only to illustrate principal agreement between surface-layer data and DNS data.

---

## Referee Comment (RC2) · Anonymous Referee #3 · 22 Dec 2018

The topic of the manuscript is of fundamental importance in the field of atmospheric turbulence. The manuscript is theoretically convincing, demosntrating a thorough understanding on the physics of turbunece, but I see two major issues that require more attention.

1. The results and conclusions presented are strongly based on DNS results, but the DNS experiments are presented very briefly: only a few sentences on lines 8-17 on page 4. This is not sufficient to convince a reader on the relevance of the experiments and the accuracy and robustness of their results. The models and the experiment setup have to be described much more in detail.

2. The conclusions of the study are based on comparison of DNS results and observations, but the observations are described even more briefly than the DNS experiments:

only lines 13-14 on page 5. It is well known that ABL observations in conditions of stable stratification are very liable to errors. In fact, in Section 4 the authors pay attention on the observational challenges. Hence, it is surprising that they do not evaluate the error sources and accuracy of the observational data that they have used themselves. The good match between the DNS results and observations does not guarantee that both are error-free. They may have same kind of biases.

Hence, a major revision is needed before I can recommend acceptance of the manuscript.

---

## Author Comment (AC2) · 2 Jan 2019

Dear Reviewer,

Thank you for careful reviewing and and friendly comments. We fully agree that readers must have clear info of DNS and observational data used in our paper. Luckily, a new paper presenting detailed info about our DNS is already in press (Mortikov et al., in press). Thanks to your comment, we are including the proper reference to the manuscript.

We recall that our paper focuses on new knowledge of dissipation rate in stably stratified turbulence. In this context, we do not write much about DNS and observational data as such. We do that *on purpose*. Otherwise some readers would overlook fac-

tual essence of our paper and comprehend it as just an empirical validation of known results. We bear in mind the interests of the majority of readers forced to look through plenty of publications.

Moreover, we think that introduction of detailed description of DNS and atmospheric data into the manuscript would, first, duplicate the info that can be easily found in already published papers (referenced in our manuscript and available to readers just by click) and, second, diverge reader's attention from the basic subject matter. Instead of repeating the already published info, we give references. The accuracy of the data and possible errors are considered in corresponding papers.

The following text has been added on page 4, line 10:

To assure accuracy of numerical simulations, we employed two DNS codes: INM-RAS, and IAP-RAS. Despite being two different codes developed separately having different spatial and temporal schemes, resolutions and statistical averaging our DNS have shown quite consistent results which can be considered as a cross-validation. For the detailed description of the numerical models used see Mortikov (2016), Mortikov et al. (in press) and Druzhinin et al. (2016).

The reference has been added on page 9, line 9:

Mortikov E.V., Glazunov A.V., Lykosov V.N.: Numerical study of plane Couette flow: turbulence statistics and the structure of pressure-strain correlations, Russian Journal of Numerical Analysis and Mathematical Modelling, 34, 2, 2019 (in press).

Please find attached the revised manuscript.

Please also note the supplement to this comment:
https://www.atmos-chem-phys-discuss.net/acp-2018-978/acp-2018-978-AC2-supplement.pdf

———————————————————————

2018.

**Supplement:**

[revised manuscript text omitted]

**5 Acknowledgements**

15 The authors acknowledge support from the Academy of Finland project ClimEco No. 314 798/799; Russian Foundation for Basic Research (Projects No. 18-05-60299, 16-05-01094 A). Analysis of data from atmospheric observation was supported from the Russian Science Foundation grant No. 17-17-01210. Post-processing of numerical data from IAP was supported by the Russian Science Foundation grant No. 15-17-20009.

[Figure]

[Figure]

Figure 1: Usual height, *z*, and vertical coordinate, *z̃*, defined by Eq. (12) characterising "absolute" and "internal" geometry of the domain, respectively. Left panel shows semi-space, *z* > 0, where *z̃* = *z*. Right panel shows the layer between two horizontal walls, 0 < *z* < *d*, where *z̃* coincides with the distance from nearest surface in its close vicinity, but essentially depends on the distances from both surfaces in central part of the domain.

[Figure]

**Figure 2: Flux Richardson number, $Ri_f$, in stable stratification versus $\tilde{z}/L$ in Couette flow or versus $z/L$ in atmospheric surface layer. Empirical data used for the calibration are obtained in two series of DNS runs employing INM-RAS code (red dots) and IAP-RAS code (blue dots). Atmospheric data are taken from Arctic coastal observatory Tiski (light grey diamonds), Black sea offshore platform (light grey squares) acoustic soundings in Kalmykia steppe (light grey stars). Dark grey dots belong to very narrow near-surface layer: $0 < \tilde{z} < 50\nu/\tau^{1/2}$. Black solid line shows Eq. (11) with conventional value of von-Karman constant, $k = 0.4$, and new empirical vale of $R_\infty = 0.2$ just obtained from the best fit of Eq. (10) to DNS data from elsewhere beyond the layer $0 < \tilde{z} < 50\nu/\tau^{1/2}$, where molecular transports are significant and Eq. (10) is not necessarily relevant.**

[Figure]

**Figure 3: Dimensionless dissipation rate in stable stratification $\varepsilon_K \tilde{z}/\tau^{3/2}$ versus $\tilde{z}/L$ in Couette flow or $\varepsilon_K z/\tau^{3/2}$ versus $z/L$ in atmospheric surface layer. Empirical data are from the same sources as in Figure 2. Black solid line shows Eq. (11) with $k = 0.4$ and $R_\infty = 0.2$. Dark grey dots belong to very narrow near-surface layer: $0 < \tilde{z} < 50\nu/\tau^{1/2}$, where molecular transports are significant and Eq. (11) is not relevant.**

[Figure]

**Figure 4: Energy Richardson number, $Ri_E = E_p/E_K$, versus $\tilde{z}/L$ in Couette flow or versus $z/L$ in atmospheric surface layer. Empirical data are from the same sources as in Figures 2 and 3. Black solid line shows Eq. (16) with $k = 0.4$, $R_\infty = 0.2$, and $C_P = 0.62$.**